# CRISPR-Cas System: A Tool to Eliminate Drug-Resistant Gram-Negative Bacteria

**DOI:** 10.3390/ph15121498

**Published:** 2022-11-30

**Authors:** Rajeshwari Kundar, Karuna Gokarn

**Affiliations:** 1Department of Microbiology, Sir H.N. Medical Research Society, Sir H.N. Reliance Foundation Hospital & Research Centre, Mumbai 400004, Maharashtra, India; 2Department of Microbiology, St. Xavier’s College, 5- Mahapalika Marg, Mumbai 400001, Maharashtra, India

**Keywords:** Gram-negative bacteria, antimicrobial resistance, CRISPR-Cas system, CRISPRi

## Abstract

Rapidly emerging drug-resistant superbugs, especially Gram-negative bacteria, pose a serious threat to healthcare systems all over the globe. Newer strategies are being developed to detect and overcome the arsenal of weapons that these bacteria possess. The development of antibiotics is time-consuming and may not provide full proof of action on evolving drug-resistant pathogens. The clustered regularly interspaced short palindromic repeats/CRISPR-associated protein (CRISPR/Cas) systems are promising in curbing drug-resistant bacteria. This review focuses on the pathogenesis of Gram-negative bacteria, emergence of antimicrobial drug resistance, and their treatment failures. It also draws attention to the present status of the CRISPR-Cas system in diagnosisand treatment of Gram-negative bacterial infections.

## 1. Introduction

### 1.1. Drug Resistance

Antibiotics have always been considered the weapon of choice to target common infections since their discovery in 1928. As the world progressed further, antibiotics gained a lot of importance in the health care system, food production, animal husbandry, and agriculture. The race between antibiotics and microorganisms has led to antibiotic resistance, one of the most important global crises. The extensive use and abuse of antibiotics, poorly controlled regulation, and inadequate surveillance has created a selective pressure that has allowed bacteria to survive and evolve into antibiotic-resistant strains [1]. An increase in drug-resistant bacteria poses a public threat that must be eradicated with newer alternatives other than antibiotics (Table 1).

Microorganisms evade antimicrobial action by employing three related mechanisms: tolerance, resistance, and persistence [1]. Resistance can be classified into two categories: intrinsic resistance and acquired resistance. Intrinsic resistance is due to inherent features such as functional or structural characteristics in which bacteria lack a target site for a particular antibiotic. Acquired resistance involves genetic mutation or acquiring genes from different bacterial strains via horizontal gene transfer leading to a drug-resistant phenotype [3]. Whereas, tolerant and/or persistent bacterial cell population/s survive prolonged antibiotic treatment because of dormancy in growth and metabolism [1,4]. Although not all groups of bacteria are resistant to antibiotics, there are six major multi-drug resistant (MDR) bacteria known as the ESKAPE bugs that evade the action of antibiotics. These ESKAPE bugs include *Enterococcus faecium*, *Staphylococcus aureus*, *Klebsiella pneumoniae*, *Acinetobacter baumannii*, *Pseudomonas aeruginosa*, and *Enterobacter* species and are responsible for nosocomial infections worldwide [5,6,7].

The spread of infections caused by Gram-negative MDR bacteria will always be a concern, due to the presence of their outer membrane lipid bilayer, which restricts entry of drugs [8,9]. Other strains apart from ESKAPE pathogens that show drug resistance include *Helicobacter pylori*, *Salmonella species*, *Campylobacter jejuni*, *Neisseria gonorrhoeae*, *Haemophilus influenzae*, and *Shigella species* [9]. Increasing resistance to antimicrobials will influence the surge of infectious diseases. In addition, it will also jeopardize antibiotic treatment of common infections associated with immunosuppression, intubation, catheterization, and other such procedures [10]. The pace of the discovery of new antibiotics is much slower than the rate of emerging drug resistance. Therefore, newer strategies are required that can tackle MDR superbugs.

### 1.2. Approaches to Overcome Drug Resistance

Current approaches against MDR bacteria include antimicrobial peptides [11,12], anti-virulence compounds [13], phage therapy [14,15], nanoparticles [16,17], drug repurposing [18,19], and vaccines [20,21]. However, there are limitations to these strategies because of which new alternatives should be explored that can help in quick diagnosis and boost treatment efficacy.

RNA-based therapeutics [RNA interference (RNAi), antisense oligonucleotides, and steric-blocking oligonucleotides] are also alternatives to treat antimicrobial drug-resistant (AMR) organisms. These therapeutics utilize oligonucleotides to enzymatically target mRNA which allows the removal of the gene that confers a resistant phenotype [22]. Antisense RNA-based technique has also provided means by which researchers can keep track of resistance-causing genes and their mutations by continual sequencing and redesigning the antisense oligonucleotides [23]. The method also provides means to identify essential genes required for growth which can be targeted [24]. However, poor intracellular uptake and chemistry-dependent toxicities pose a major disadvantage for use of RNA as therapeutics [22].

Gene editing tools that employ restriction enzymes such as zinc finger nucleases (ZFN) and trans-activator-like effector nucleases (TALENS) can be tailored and designed to cleave specific sequences of DNA [25]. Both these enzymes have their cleavage domain fused to a custom-made DNA binding domain which allows them to bind and cleave the DNA sequence [26]. ZFN and TALENS editing tools have thus paved a path for the rise of clustered regularly interspaced short palindromic repeats/CRISPR-associated proteins (CRISPR/Cas)-based diagnosis as well as therapeutics (Figure 1).

CRISPR-Cas system is the most extensively studied genome editing technology in the modern era. This technology is quick, less expensive, and is an efficient gene editing tool that has been shown to improve genetic defects [27,28], remove bacterial pathogens [29,30], and has eliminated major infectious viruses [31,32]. Many scientific studies have shown to control the spread of antibiotic resistance via the use of the CRISPR-Cas-based approach [32,33,34].

This review article focuses on Gram-negative bacterial drug resistance and the use of the CRISPR-Cas system for diagnosing and treating infections caused by these bacteria. The topics covered in this review are as follows:Gram-negative bacteria and pathogenesis;Drug resistance in Gram-negative bacteria;Failures in recent therapies;CRISPR-Cas system as a tool
In diagnosis to detect drug-resistant and other pathogenic bacteria;In treatment to eliminate AMR Gram-negative pathogenic bacteria;Challenges faced.

## 2. Gram-Negative Bacteria and Pathogenesis

Gram-negative bacteria show the presence of an envelope [35,36,37] that is made up of three layers—(a) The outer membrane is a lipid bilayer made up of lipopolysaccharide (LPS), an endotoxinat its outer leaflet and phospholipids at the inner leaflet. It also contains external and transmembrane proteins which allow the entry of small hydrophilic molecules. This layer acts as a protective barrier against antibiotics and other harmful compounds in Gram-negative bacteria. (b) The thin peptidoglycan layer consists of repeating units of the disaccharide N-acetyl glucosamine and N-acetyl muramic acid. (c) The inner membrane encloses the cytoplasm and is responsible for various bacterial cell functions such as transport, biogenesis, and structure.

### 2.1. Pathogenesis

The incidence of Gram-negative bacterial resistance responsible for both hospital-acquired and community-acquired infections is majorly due to harbouring of resistance genes on extra-chromosomal DNA. Most of them belong to the Enterobacteriaceae family and non-fermenting Gram-negative bacteria. Medically important Gram-negative bacteria include *Acinetobacter* spp., *Bordetella pertussis*, *Campylobacter* spp., Enterobacteriaceae (*Citrobacter* spp., *Enterobacter* spp., *Escherichia coli*, *Klebsiella* spp., *Salmonella* spp., *Serratia marcescens*, *Shigella* spp., *Yersinia* spp.), *Acinetobacter* spp., *Bordetella pertussis*, *Campylobacter* spp., *Haemophilus influenzae*, *Helicobacter pylori*, *Legionella pneumophila*, *Neisseria* spp., *Pseudomonas aeruginosa*, and *Vibrio cholerae* [38,39].

#### 2.1.1. Hospital-Acquired Infections (HAI)

HAI are of major concern to patients in healthcare facilities. These infections are majorly associated with invasive medical devices and surgical procedures. Infections caused by Gram-negative bacteria contribute to more than 30% of the total HAI [40]. Indeed, infections caused by Gram-negative bacteria are responsible for 45–70% of ventilator-associated pneumonia (VAP), 20–30% of catheter-related bloodstream infections, and other intensive care units (ICU)-related infections involving sepsis at the surgical site. Microbes responsible for HAI include Enterobacteriaceae and the non-fermenting Gram-negative bacteria [41]. Almost 80% of urinary tract infections (UTI) are caused by *E. coli* out of which 30% of the infections are catheter-associated [42]. Hospital-acquired pneumonia is caused by a variety of Gram-negative pathogens such as *P. aeruginosa* (21.8%), *Klebsiella pnemoniae.* (9.8%), *E. coli* (6.9%), *A. baumannii.* (6.8%), and *Enterobacter* spp. (6.3%) [43]. *K. pneumoniae* carbapenemases cause pneumonia, bloodstream infections [44], nosocomial infections [45], and neonatal infections [46]. *A. baumannii* causes nosocomial infections by forming biofilms on inanimate objects, thus giving rise to various hospital-acquired outbreaks such as meningitis, bacteremia, VAP, UTI, and other wound infections [47]. *Enterobacter* spp. caused 4.7% of infections in the ICU and is found to be the fifth most common pathogen isolated from the ICU. Nosocomial infections caused by *Enterobacter* include bacteremia, respiratory infections, UTI, endocarditis, soft tissue infections, and osteomyelitis [48].

#### 2.1.2. Community-Acquired Infections

Gram-negative bacteria are also responsible for community-acquired infections. These include urinary tract infections (UTI) caused by *E. coli* (66.6%) and other Enterobacteriaceae spp. such as *K. pneumoniae* (16.6%) and *Proteus mirabilis* [49]. *E. coli* is also found to cause community-acquired pneumonia with a higher mortality rate than pneumococcal pneumonia [50]. *K. pneumoniae* infections comprise 4.8% of the common causative bacteria of community-acquired pneumonia [51]. *Neisseria gonorrhoeae*, a Gram-negative coccus is also found to be the cause of community-acquired sexually transmitted disease [52]. *Burkholderia cepacia*, an opportunistic pathogen is responsible for causing ‘swamp rot’ a foot infection in patients with cystic fibrosis [53].

### 2.2. Drug Resistance Mechanism in Gram-Negative Bacteria

Gram-negative bacteria are either multi-drug resistant (MDR), extensive drug-resistant (XDR), or pan-drug resistant (PDR). MDR bacteria are those that are resistant to multiple antibiotics, classes, or subclasses of antibiotics [54]. For example, MDR bacteria such as the carbapenem-resistant Enterobacterales (CRE) show resistance to a class of beta-lactam antibiotics such as carbapenem [55]. XDR-bacteria show drug resistance to multiple antibiotics which come under the most standard antibiotic regimen [54]. XDR pathogens include the XDR strains of *P. aeruginosa* which are found to be susceptible to only one or two classes of drugs used to treat *Pseudomonas* infections. Similarly, XDR strains of *A. baumannii* are found to have resistance to all carbapenem drugs and are opportunistic pathogens that cause major outbreaks in healthcare settings [56]. PDR bacteria show drug resistance to all the commercially available antibiotics [54]. PDR strains of *A. baumannii* causing nosocomial infections are resistant to all antibiotics, including the drug colistin which is the last resort to treat infections caused by MDR pathogens [57].

#### Mechanisms of Antibiotic Resistance

Gram-negative bacteria show all three major kinds of resistance mechanisms (as shown in Figure 2) which include intrinsic, acquired, and adaptive resistance.

Intrinsic resistance enables bacteria to avoid the antimicrobial activity of drugs. This kind of resistance solely depends on the genome of the bacteria. Intrinsic resistance can be because of a lack of a target site for the drug, low permeability for the drug, chromosome-based expression of efflux pump, or drug inactivation [59].

*P. aeruginosa*, an opportunistic pathogen shows intrinsic resistance to various antibiotics due to the expression of two efflux systems MexAB-OprM and MexXY-OprM [60]. Consequently, the level of resistance in *P. aeruginosa* is further increased because of a modified OM expressing different porin channels and the production of drug-inactivating enzymes such as beta-lactamases and aminoglycoside-modifying enzymes [61]. Species of *Burkholderia* are naturally resistant to polymyxin B and antimicrobial peptides because of their modified OM, expression of metalloproteases, and presence of an efflux system. Some members of the genus *Burkholderia* are also able to survive in the presence of chlorhexidine in hospital disinfectant solution and some can grow using penicillin as their carbon source [62].

Some Gram-negative bacteria also show resistance to the antibiotic by having a low affinity to the drug or lacking the drug binding site. Randall et al., demonstrated this phenomenon of intrinsic resistance in Gram-negative pathogens such as *P. aeruginosa* PAO1, *E. cloacae* ATCC 13047, *K. pneumoniae*, *Moraxella catarrhalis*, and *S. typhimurium* that were resistant to daptomycin. This is because daptomycin requires a substantial proportion of anionic phospholipids for their calcium-dependent insertion into the cell membrane. Gram-negative bacteria contain an insufficient amount of anionic phospholipids due to which they are resistant to the effects of daptomycin [63]. Drug-resistant *Pseudomonas* may lack target sites and therefore are not susceptible to the action of the drug, for example, triclosan as they lack the target site for the biocide [64].

Adaptive resistance is a transient increase in the ability of a bacterium to survive in the presence of an antibiotic because of genetic alteration triggered by environmental stimuli. The resistance arises due to the interaction of bacteria and a drug [65]. Adaptive resistance is dependent on the environmental signals (such as anaerobiosis, pH, presence of ions, polyamines, carbon sources, and exposure to nonlethal doses of antibiotics) that may be reverted after the removal of the stimulus. *P. aeruginosa* is a commonly found opportunistic pathogen in the lungs of patients suffering from cystic fibrosis (CF). They can survive in the airways of the lungs of patients suffering from CF by multiple mechanisms. Because of the slightly anaerobic environment (due to the accumulation of mucus), these bacteria perform microaerobic respiration by expressing higher affinity terminal oxidases [66]. Interestingly, *P. aeruginosa* is also found to be hypermutable in CF patients. They can turn on/off their gene expression depending on their environmental condition. In a hypoxic environment, certain *P. aeruginosa* mutants secrete exopolysaccharides and form mucoid colonies which allow them to survive in the presence of antibiotics and protect them from mucociliary clearance [67]. *Pseudomonas* can use polyamines as a source of carbon and nitrogen. In a research study by Kwon and Lu, it was found that in the presence of polyamines such as spermine and spermidine, the minimum inhibitory concentration (MIC) of ciprofloxacin and polymyxin B for clinical isolates of *P. aeruginosa* was higher. Moreover, the *P. aeruginosa* PAO1 strain showed resistance to the beta-lactam imipenem in the presence of polyamines which blocked the porin channels and reduced the membrane permeability of the drug [68]. *Salmonella enterica* serovar Typhimurium shows adaptive resistance to polymyxin B due to the presence of low magnesium ions in the environment. PhoP-PhoQ and PmrA-PmrB are two-component signaling systems present in the *Salmonella enterica* strain that activate genes of the *pmr* operon on detecting low Mg^+2^ levels via the pmrD protein. This leads to the activation of genes that modify the LPS layer of the bacteria which reduces the negative charge of the OM. As a result of this, positively charged antibiotic interaction with the OM is reduced [68,69,70].

Another example of adaptive resistance is the study by Kang and Seo, who demonstrated that on exposure to acid stress and combined salt and acid stress, *Salmonella enteritidis* showed increased resistance to the drug ciprofloxacin and erythromycin. After prolonged refrigeration, *E. coli* O157:H7 exhibited an increase in resistance to certain antibiotics belonging to the family of beta-lactam antibiotics, macrolides, aminoglycosides, and tetracyclines [71].

Acquired resistance in bacteria is due to the acquisition of genes that confer a resistant phenotype. In this kind of resistance, the bacteria gain the ability to survive and grow in the presence of antibiotics to which it was earlier susceptible. Acquired resistance can be either due to mutation and selection or by the acquisition of genes through genetic mechanisms that include conjugation, transformation, or transduction [72]. These genetic mechanisms can give rise to three major resistance mechanisms which include reduced drug uptake or high efflux activity, inactivation of antibiotics via enzymatic modification or degradation of antibiotics, and lastly modification of the drug binding site or overproduction of the target protein [58,73]. Strains of *E. coli* showing resistance to the antibiotic cefoxitin showed an absence of OmpF and OmpC porins [74]. Similarly, a mutation in the outer membrane channel (OMC) gene OprD has led to the loss of D2 porin in *P. aeruginosa* and the loss of CarO porin in *A. baumannii* has given rise to imipenem resistance [75]. Increased MIC for imipenem was also shown in *K. pneumoniae* isolates that lacked OmpK35/OmpK36 porins [76]. Efflux pumps in bacteria are categorized into five distinct families. All five families of efflux pumps are found in Gram-negative bacteria. These families of efflux pumps include the ABC transporter family, MATE transporter family, SMR transporter family, MFS transporter family, and the RND transporter family (exclusive in Gram-negative bacteria) [77]. VcaM is an ABC type of MDR transporter that provides resistance in *V. cholerae* to tetracycline, norfloxacin, ciprofloxacin, and others [78]. The AcrAB-TolC pump belonging to the RND family of transporters confers resistance in *E. coli* against chloramphenicol, tetracyclines, macrolides, and many more antibiotics [77]. Isolates of *E. coli*, *P. aeruginosa*, *K. pneumoniae*, and *A. baumannii* are resistant to all beta-lactam antibiotics due to the expression of extended-spectrum beta-lactamases (ESBL), *K. pneumoniae* carbapenemases (KPC), oxacillinase (OXA), and carbapenemases that degrade the antibiotic. Similarly, the inactivation of drugs using chemical groups for modification is seen in the *Campylobacter coli* strain that encodes six aminoglycosides-modifying enzymes that confer resistance to various aminoglycosides used in the treatment of *Campylobacter* infections [79]. Mutations in the *gyrA* gene encoding DNA gyrase, a target site for the drug fluoroquinolones have led to fluoroquinolone resistance in *E. coli* and other Gram-negative bacteria due to lower affinity for the drug [80]. Table 2 summarizes the drug resistance in Enterobacteriaceae, *P. aeruginosa*, and *A. baumannii*.

Thus, Gram-negative bacterial OM selectively keeps drugs from entering. Drug-resistant Gram-negative bacteria have pumps to efflux important antibiotic drugs. In addition, they destroy antibiotics enzymatically or have receptors that have undergone a mutation so that the drug is unable to bind. They also develop new processes that can bypass the effects of drugs.

## 3. Failures in Recent Approaches to Treat AMR in Bacteria

Newer approaches and alternate therapies are arising on a larger scale to defeat antimicrobial resistance. These alternatives employ numerous strategies to curb resistance via the use of combinatorial therapy, nanoparticles, phage therapy, and antibodies [81]. Although these approaches have shown promising results, there are many challenges.

Combinatorial therapies involve the use of a combination of drugs rather than a single drug to produce a synergistic effect that can attack multiple targets and kill target bacteria. However, the usage of more than a single drug in patients carrying drug-resistant pathogens might lead to incompatibility among antibacterial agents, affect the pharmacodynamics and pharmacokinetics interaction and may also increase the carriage and transmission of drug-resistant pathogens [82,83].

Nanoparticles have shown promising results in targeting major drug-resistant bacteria such as vancomycin-resistant enterococci (VRE), MDR *P. aeruginosa*, *S. aureus*, and MDR *A. baumannii.* However, nanoparticles show limitations that hinder their use in clinical settings. This is due to the accumulation of these particles in different organs of the body, and limited knowledge is available about their biocompatibility and interactions with healthy mammalian cells, moderate stability, and cytotoxic effects on the body [83,84].

Phage therapy is another approach that has proven to be effective against AMR pathogens [85]. However, some factors pose a challenge in the use of phage therapy to treat bacterial infections. Drawbacks involved in phage therapy include their use against intracellular bacteria, the production of neutralizing antibodies against the phage, and the emergence of phage resistance in bacteria [86].

Lastly, monoclonal antibodies have also emerged as one of the methods to treat bacterial drug resistance. Some barriers that impede the use of monoclonal antibodies are [87]:Bacterial target selection (e.g., LPS have many serotypes),Ineffectiveness of a single monoclonal antibody to treat a complex bacterial infection,Degradation of antibodies by bacterial proteolytic enzymes.

Therefore, extensive research is required to find other alternatives that can overcome the limitations and challenges of the approaches. The CRISPR-Cas system is one such emerging platform that has been employed in the detection and treatment of bacterial pathogens.

## 4. CRISPR-Cas System to Overcome Drug-Resistance

### 4.1. Introduction to CRISPR-Cas System

Prokaryotic organisms contain two main classes of short sequence repeats known as the continuous repeat and the interspersed repeat. Bacteria and archaea contain a group of interspersed repeats with a unit length of less than 200 base pairs (bp) that are non-protein coding, intercistronic, and widely distributed throughout their genome. These repeats belong to a family of repeats called the clustered regularly interspaced short palindromic repeats (CRISPR) [88]. The CRISPR loci are diverse. It is made up of direct repeats consisting of 21–48 bp alternate with non-repetitive sequences of 26–72 bp known as the spacer sequences [89]. Upstream of the CRISPR locus is several hundred base pair sequences containing larger repeats flanking the locus called the leader sequence. These sequences are rich in adenine and thymine and are non-coding as they lack an open reading frame (ORF) [90]. Transcription of the CRISPR locus begins from a promoter region which lies in this leader sequence [89]. The repeats in CRISPR loci have dyad symmetry with a stem-loop-like structure at the termini because of the presence of complementary sequences GTT and AAC [90].

Adjacent to the CRISPR locus is an operon of CRISPR-associated genes called the *cas* genes. These *cas* genes are present only in prokaryotes that have CRISPR loci. There are four core *cas* genes namely *cas1*, *cas2*, *cas3*, and *cas4* arranged in *cas3*, *cas4*, *cas2*, *cas1* alignment near the CRISPR locus [91]. Cas1 and Cas2 proteins function as nucleases that participate in spacer acquisition and are found in all species of prokaryotes having the CRISPR loci [92]. Cas3 protein performs the role of the helicase and Cas4 acts as a DNA-binding exonuclease [91]. There are 40 *cas* genes identified that encode a heterogeneous family of Cas proteins with the functional domain of polymerases, various RNA-binding proteins, helicases, and several nucleases [88,93].

### 4.2. Mechanism and Role of CRISPR-Cas in Adaptive Resistance in Prokaryotes

Bacteriophages are present abundantly in the environment. Microorganisms have devised various schemes to target these phages along with the other foreign DNA present in the environment. CRISPR is found to be one such strategy that microbes employ against viruses and plasmids (Figure 3).

The evidence of the role of CRISPR in adaptive immunity came from *Streptococcus thermophilus* which showed resistance to bacteriophages by acquiring the phage sequence into the CRISPR loci [89,94]. When a bacteriophage or a foreign nucleic acid invades a bacterial cell containing the CRISPR Cas system, it uses the CRISPR-based mechanism to counteract the attack by the invading DNA. This mechanism provides a CRISPR-based adaptive immunity against foreign DNA and phages upon re-infection into the same cell and the bacterial cell becomes resistant to them [95].

The three steps involved in CRISPR-based immunity include:Adaptation—The adaptation step is conducted by the Cas1 and Cas2 proteins with the help of other effector proteins. In this step, the exogenous DNA is cleaved, followed by the recognition of the proto-spacer adjacent motif (PAM) that consists of type-specific short sequences (2–3 nucleotides) for the selection of the proto-spacer. This proto-spacer is then processed into a pre-spacer having the last PAM nucleotide. The leader end repeat sequence in the CRISPR locus is cleaved following which the pre-spacer is integrated along with duplication of the repeats flanking the spacer [96,97].Expression—This step involves the biogenesis of the CRISPR RNA (crRNA) by the transcription of the CRISPR locus containing the spacer sequence. This process occurs when the bacterial cell is re-infected with the same phage or foreign DNA. First, the primary CRISPR transcript called the pre-crRNA is generated. These transcripts are further processed by different proteins such as Cas5, Cas6, or RNase III depending on the type of CRISPR system (I, II, III) involved. Modification of the pre-crRNA process gives rise to a mature crRNA that contains spacer sequences flanked by partial repeats [98,99].Interference—In this step, the mature crRNA complexed with the Cas nucleases guides the Cas machinery to the complementary sequences present on the invading nucleic acid. The binding of the crRNA to the homologous sequences leads to the cleavage of the foreign DNA [100,101].

### 4.3. Types of CRISPR Cas System

The CRISPR-Cas system is broadly classified into two classes, the Class 1 system and the Class 2 system based on their signature genes and the organization of the CRISPR loci. Class 1 consists of types I, III, and IV along with 16 subtypes. In the Class 1 CRISPR Cas system, the multi-subunit effector Cas proteins are complexed with the crRNA which together conduct the processing and interference mechanism. Class 2 contains the types II, V, and VI together with 17 subtypes. This class includes a single, large effector protein with multiple domains complexed with the crRNA which conducts all the functions. These multi-domain proteins include Cas9 in type II, Cas12 in type V, and Cas13 in type VI [102,103]. The characterization of each type is mentioned in Table 3.

## 5. Applications of CRISPR Cas System

CRISPR Cas system is based on the mechanism of RNA-guided Cas proteins that allow the cleavage of specific targeted DNA. The type II CRISPR Cas system is most widely used for its various genome engineering application. This is because Type I and III employ multiple effector Cas proteins for the cleavage of the foreign DNA whereas the type II system uses only a single multi-domain protein Cas9 for RNA-guided cleavage of DNA [101]. The CRISPR Cas9 system is characterized by the Cas9 protein endonuclease, which requires processing the pre-crRNA via transactivating RNA (tracrRNA) encoded upstream of the type II system in *Streptococcus pyogenes.* This binding activates endogenous RNase III which processes the pre-crRNA to give rise to tracrRNA: crRNA. This RNA duplex complexes with the Cas9 protein to form dual RNA: Cas9 which is then directed for the interference mechanism [108,109]. This system has gained scientific interest because of the ease by which it can be reprogrammed in other living systems. The challenges in treating drug-resistant infections have led us to consider the gene-editing CRISPR Cas9 system as a potential antibacterial that uses different sgRNA that can guide the efficient removal of desired DNA sequences [110].

### 5.1. In Diagnosis

#### CRISPR Cas9 Tool for Detection of Pathogenic Gram-Negative Bacteria and AMR in Gram-Negative Bacteria

The CRISPR Cas system has been extensively studied for its use in diagnostics as a detector system. Modifications in CRISPR systems have allowed us to improve the specificity as well as the cost efficiency of traditional tools used in diagnoses such as polymerase chain reaction (PCR), isothermal-based amplification assays and sequencing. Next-generation detection tools that are single nucleotide specific are important for the identification of variation in dangerous pathogens as well as for the detection of mutations to improve drug therapies and overcome AMR. Recently designed detection tools modified using CRISPR-based systems majorly detect nucleic acid present in the sample [111]. Newly developed CRISPR Cas-based detection tools carry out their detection process either using the Cas9 enzyme which allows the detection of a pathogen by specific binding and cleavage of target DNA or by using other classes of Cas proteins that detect pathogen by collateral cleavage.

Various detection systems involving the use of amplification methods combined with CRISPR/ Cas9 are developed for the detection of different pathogens [111,112]. These recently developed tools employing Cas9 include CRISPR/Cas9-triggered isothermal exponential amplification reaction (CAS-EXPAR) system [113], CRISPR-typing PCR (ctPCR), and CRISPR-associated reverse PCR (CARP) [114], CRISPR/Cas9-mediated lateral flow nucleic acid assay (CASLFA) [115], CRISPR-mediated DNA-fluorescence in situ hybridization (FISH) [116], finding low abundance sequences by hybridization (FLASH)—next generation sequencing (NGS) [109], etc. The variant form of Cas9 called the Cas9 nickase that performs only single-stranded break is also used in a few detection tools such as Cas9nAR (Cas9 nickase-based amplification reaction) [117], and the Paired dCas9 (PC) reporter system for the detection of Gram-negative bacteria [118].

However, very few systems involving the Cas9 enzyme are developed which can specifically detect Gram-negative bacteria and AMR genes associated with them. Sun et al., detected the presence of *E. coli* O157:H7 using CRISPR Cas9 that cleaved the target sequence which triggered strand displacement amplification followed by rolling circle amplification. The detection was done by quenching the fluorescent probes that hybridized with the replication products based on a metal-organic framework platform and the fluorescence recovery was measured [119]. Kim et al., reported a CRISPR-mediated surface-enhanced Raman scattering (SERS) assay for the detection of MDR bacteria. The assay combined the individual activity of CRISPR dCas9, surface-enhanced Raman scattering (SERS), and the separation property of magnetic nanoparticles to detect three MDR bacteria including *S. aureus*, *A. baumannii*, and *K. pneumoniae.* They also demonstrated the on-site capture and detection of the MDR pathogen using this assay [120].

The Cas9nAR system has been used for the detection of *S. typhimurium* by targeting the *invA* gene and the *uidA* gene of *E. coli*. This system relies on the action of sgRNA: Cas9 nickase complex which creates a single-stranded break at the target DNA sequence. Following this, Exo’ Klenow polymerase extends the nicked strand with the help of primer 1 to produce a complementary DNA sequence and the ssDNA overhangs are displaced. Primer 2 then binds to the displaced ssDNA and begins extension. Priming and extension of the DNA strands again activate the Cas9 nickase to cleave the target DNA sequence extended and hence, the cycle gets repeated. Real-time monitoring of the Cas9 nickase activity using SYBR Green I show a fluorescence intensity directly proportional to the concentration of dsDNA products [117].

Apart from the Cas9 enzymes, Cas proteins belonging to other types have also been used in the detection system. The Cas12a and its orthologues (LbCas12a, LbaCas12a from *Lachnspiraceace* bacterium ND2006 and FnCas12a from *Francisella novicida* U112) have been used most widely for bacterial detection followed by the Cas13a and its orthologues (LbuCas13a from *Leptotrichia buccalis*, LwaCas13a and LwCas13a from *Leptotrichia wadei*) [121]. Qui et al., used the CRISPR Cas12a technology to develop the CRISPR-HP assay for the detection of *H. pylori* from stool specimens. The assay involved three major steps involving recombinase polymerase amplification followed by the CRISPR Cas12a reaction and lastly detection using lateral flow biosensor [122]. Interestingly, Wu et al., developed a polypropylene (PP) bag-based CRISPR Cas12a method to detect the presence of *S. typhimurium* at home in a three-chambered PP bag that allows the lysis, washing and isothermal amplification/detection following the addition of nucleic acids [123]. The CRISPR Cas12a and the multienzyme isothermal rapid amplification (MIRA) system were used to develop a rapid nucleic acid detection tool for detecting *E. coli* O157: H7 from food samples [124]. Lee et al., also used the CRISPR Cas12a combined with the loop-mediated isothermal amplification (LAMP) technique to develop a rapid, sensitive, and visualizing method for the detection of *E. coli* O157: H7 from fresh vegetables [125].

CRISPR Cas12a has also been used for the detection of *Vibrio* species. Zhang et al., developed a simple, specific, and contamination-free method to conduct on-site detection of *V. parahaemolyticus* from seafood samples. The method involves DNA extraction, PCR, and CRISPR Cas12a-assisted detection of the target bacteria in the presence of a fluorophore-quencher labelled reporter [126]. Likewise, Wu et al., designed a reversible valve-assisted chip-based method for the detection of *V. parahaemolyticus* from a sample by employing the LAMP assay technique along with the CRISPR Cas12a system for its detection [127]. Xiao et al., also developed a rapid and sensitive tool called the recombinase-aided amplification (RAA) -CRISPR/Cas12a assay for the detection of *V. vulnificus*. The assay consists of DNA extraction, followed by amplification of the targeted sequence. Amplified dsDNA activates the collateral cleavage activity of Cas12a and allows pathogen detection by the fluorophore-quencher reporter system [128].

Other orthologues of Cas12 such as LbaCas12a, LbCas12a, and AapCas12b (from *Alicyclobacillus acidiphilus*) have also been used extensively for developing new bacterial detection tools. For example, the LbaCas12a was used to develop a CRISPR Cas12a-based lateral flow biosensor that could detect as low as a single copy of the acyltransferase gene of *P. aeruginosa* from clinical samples in combination with loop-mediated isothermal amplification in the presence of a reporter [129]. Sheng et al., developed MXene coupled CRISPR Cas12a system using the LbaCas12a enzyme. This system was able to detect and quantify LPS and Gram-negative bacteria both present in the sample. LPS is detected by an aptamer-based system that in its presence prevents the activity of LbaCas12a whereas the target bacteria are detected by the trans-cleavage activity of LbaCas12a in the presence of a reporter [130]. Bonini et al., used the collateral activity of LbaCas12a coupled with electrochemical impedance spectroscopy (EIS) to enable a label-free bio-sensing assay for the detection of *E. coli* and *S. aureus*. The presence of the target sequence activates the LbaCas12a activity which cleaves ssDNA embedded on a sensor surface, reducing the charge transfer resistance detected by electrical output [131]. The CRISPR Cas12a system was also able to detect more than one drug-resistant gene simultaneously with high accuracy. This was demonstrated by Wang et al., by establishing a rapid multiplexing method for the detection of MDR genes in *A. baumannii* with high specificity by using a synergistic combination of multiplex PCR and CRISPR LbaCas12a [132].

To increase the sensitivity and ease of detection, Cai et al., designed single digit *Salmonella typhimurium* detection device by incorporating bio-barcode immunoassay (BCA), recombinase polymerase amplification (RPA), and CRISPR-Cas12a cleavage in a single reaction system to allow the sensitive and visual detection of *Salmonella* from the sample [133]. Another orthologue of Cas12a has also been used such as the LbCas12a. This Cas12a orthologue was used by Yin et al., to develop a G-quadruplex-based CRISPR-Cas12a bioassay for the ultrasensitive detection of bacteria and was successfully able to detect *Salmonella* from the sample [134]. Similarly, other researchers have also tried different approaches to find interesting applications of LbCas12a such as the detection of *E. coli* O157:H7 by the one-pot toolbox with precision and ultra-sensitivity (OCTOPUS) platform [135], detection of *Yersinia pestis* using the Cas12a- up-converting phosphor technology (UPT)-based LFA (UPT–LFA) assay [136], and detection of foodborne pathogens such as *E. coli*, and *V. parahaemolyticus* using the recombinase polymerase amplification with CRISPR-Cas12a for food safety (termed RPA-Cas12a-FS) method [137].

Another type of Cas protein employed in the detection of the pathogen is the CRISPR Cas13a which performs RNA-mediated RNA cleavage. However, the orthologue of Cas13a has been used for the detection of Gram-negative bacteria such as the LbuCas13a-based detection tool called allosteric probe-initiated catalysis and CRISPR-Cas13a (APC-Cas) was used in the detection of *Salmonella enterica* serotype Enteritidis from the sample. The detection tool is a DNA extraction-free strategy called allosteric probe-initiated catalysis and CRISPR-Cas13a (APC-Cas) system which uses pathogen directly present in the sample. The system relies on an aptamer containing a T7 promoter sequence. In the presence of the pathogen, the aptamer binds to its surface followed by the binding of the primer to the aptamer. DNA polymerase then catalyzes DNA synthesis, and the pathogen is released. Following this, RNA synthesis is initiated with the addition of T7 polymerase. RNA molecules produced are bound by the Cas13a-crRNA which on activation performs trans-cleavage (non-specific cleavage) of the non-specific reporter ssDNA resulting in a fluorescence signal that allows the detection of the pathogen [138]. On a similar basis, Gao et al., used the LwCas13a to develop a PCR-CRISPR-Fluorescence (PCF) based nucleic acid detection for the detection of *Salmonella* spp. from the sample sensitively and specifically [139].

Thus, different modifications in traditional methods when synergistically combined with CRISPR Cas system components were found to have high accuracy and sensitivity and are cost as well as time effective. These detection systems can pave the path for the early detection of harmful bacteria that may be present in different samples. Along with the nucleic acid-based detection system that allows us to detect the presence of certain genes, there is also a need to develop detection tools that will enable us to detect the presence of a pathogen in the sample based on its unique cell components such as LPS, endotoxins, and exotoxins.

### 5.2. Treatment of AMR Gram-Negative Bacteria

The CRISPR-Cas system can overcome the problems caused by drug-resistant bacteria by specifically targeting genes responsible for AMR. It can be used to target both chromosomal-encoded genes and plasmid-encoded genes. Depending on the location of the target gene, two different approaches can be considered using the CRISPR Cas system as an antibacterial agent. These approaches are pathogen-focused and gene-focused [140].
In the pathogen-focused approach, chromosomal genes are targeted which results in the death of the bacteria. This approach can be used in the treatment of specific infectious diseases as the CRISPR Cas system will selectively eliminate the disease-causing bacteria from the microbial community.In the gene-focused approach, the plasmid-encoded genes either responsible for the plasmid replication or drug resistance are targeted. This approach using the CRISPR Cas system will help eliminate AMR genes from the bacteria or result in plasmid curing when plasmid replicons are targeted. As a result, the bacteria will become sensitive to antibiotics and the chance of plasmid transfer between bacterial species will be reduced.

Ironically, the CRISPR Cas system is a well-studied bacterial defense system that has been exploited and reprogrammed to kill itself. This has been proved in experiments conducted against certain Gram-negative bacteria by using the pathogen-focused approach. This was demonstrated by Citorik et al., who used the CRISPR Cas system to target the virulence factor *eae* gene in *E. coli* O157:H7 and determined the reduction in viability of the bacteria as the gene is essential for colonization and pathology [141]. Gomaa et al., and his colleagues also employed the pathogen-focused approach by using the Type I CRISPR Cas system for the specific removal of *E. coli* strains and *S. enterica* from mixed cultures by targeting chromosomal genes required for metabolism and cell division [142]. Hamilton et al., conducted a similar experiment by using a plasmid encoding conjugation and CRISPR systems both such that it targeted the specific killing of *S. enterica* in biofilms due to the activity of Cas9 and high conjugation efficiency. Such conjugative systems could serve as an ideal antibacterial to clear biofilms and prevent the rise in drug resistance [143].

Furthermore, interesting experiments performed by Kiga et al., and his colleagues showed efficient bactericidal activity of CRISPR Cas13a against *E. coli* strains carrying genes for carbapenem (*bla*_IMP-1_, *bla*_OXA-48_, *bla*_VIM-2_, *bla*_NDM-1_, and *bla*_KPC-2_) and colistin (*mcr-1* and *mcr-2*) resistance on a plasmid and/or chromosome, respectively. Their study showed that reprogramming and delivery of the CRISPR Cas13a using M13 phages to target drug-resistant genes activated the non-specific RNase activity of LshCas13a (from *Leptotrichia shahii*) which resulted in cell death [144]. Similarly, Song et al., developed a *trans*-conjugative delivery system called the CRISPR Cas13a-based killing plasmids (CKPs) which targeted the endogenous transcripts of *S. enterica* serovar Typhimurium. This system when delivered in vitro using donor *E. coli* strain to *S. typhimurium* strains in mixed microflora, the CRISPR Cas13a exhibited bactericidal effects specifically against *S. typhimurium*. Furthermore, in the mouse infection model, related results proved that delivery of CRISPR Cas13a significantly reduced the colonization of *S. typhimurium* in the intestinal tract. Such an approach of using Cas13a to target RNA transcripts can be explored further to target strains of pathogenic bacteria based on the presence of specific genes encoding virulence factors [145]. However, when using the pathogen-focused approach for targeting bacterial strains other possibilities of gene mutation, in vivo efficacy, delivery in a complex microbial community, and conjugation efficacy in biofilm matrix, should be considered and studied, confirming that only pathogenic bacteria are targeted and no other microflora in the system is affected.

Considering the gene-focused approach, many in vitro studies have been conducted to demonstrate the elimination of AMR genes followed by the re-sensitization of the bacteria to antibiotics. Kim et al., demonstrated the re-sensitization of *E. coli* strains carrying plasmids encoding for ESBL. They transformed ESBL-producing *E. coli* strains with plasmids encoding for Cas9 and crRNAs against conserved regions in the ESBL genes. Successful transformation and expression of the CRISPR Cas9 system resulted in targeted cleavage of the resistant plasmids which disarmed the resistance of the *E. coli* strain [146]. Using a similar approach, Wan et al., showed a reversal of resistance in colistin-resistant *E. coli* strains. It was observed that the *E. coli* strains harboring the *mcr-1* gene when engineered with CRISPR Cas9 plasmid led to the elimination of the *mcr-1* gene. Consequently, these now susceptible *E. coli* strains also prevented horizontal gene transfer after transformation with CRISPR Cas9 plasmid [147].

The CRISPR-Cas system is also used for curing plasmids that confer resistant phenotype. This was demonstrated by Hao et al., by targeting genes involved in plasmid replication, partitioning, and encoding for carbapenemases using the CRISPR-Cas system that resulted in the curing of antibiotic-resistant genes from CRE bacteria [148]. Similarly, Yosef et al., and his colleagues showed that bacteria lysogenic with lambda phages carrying the CRISPR-Cas system as their genetic material showed plasmid curing and thus can specifically be used to prevent horizontal gene transfer as well as target resistance genes to prevent AMR [149]. When using such approaches, it is also necessary to consider that targeting genes responsible for plasmid replication can also prove to help eliminate drug-resistant genes carrying plasmids. This was proved by a study carried out by He et al., who successfully sensitized clinical isolates of *E. coli* to antibiotics in vitro by incorporating IS26-based CRISPR/Cas9 system to target both plasmid replication genes as well as antibiotic resistance genes *mcr-1, bla*_KPC-2_, and *bla*_NDM-5_ [150].

*Another strategy that is employed to target MDR genes is the CRISPR interference (CRISPRi) system. This system employs the use of an inactivated form of Cas9 called the dead Cas9 (dCas9) fused to a repressor domain*—Krüppel associated box *(KRAB). The dCas9 lacks the endonuclease activity found in Cas9. The CRISPRi system is used to knock down the gene of interest rather than gene knockout and thus can be reversed. When the dCas9 and single guide RNA (sgRNA) are expressed in the presence of an inducer, the sgRNA has a guide sequence domain which is complementary to the target sequence and guides the Cas9 protein to the target sequence for cleavage. It is a single RNA molecule that contains the short crRNA sequence fused to the scaffold tracrRNA sequence. The dCas9: sgRNA complex binds to the target sequence on the DNA and prevents the transcription initiation or elongation process. This leads to silencing the gene of interest* [151,152]. *Li et al. developed a CRISPRi system to target the class I integron in E. coli C600 that participates in antimicrobial resistance. They demonstrated the CRISPRi-mediated silencing of* IntI1 integrase-mediated integration of antimicrobial resistance gene cassettes by targeting the *intI1* gene. Subsequently, knocking down a particular sequence in the *intI1* gene resulted in lower levels of transcription of trimethoprim and sulfamethoxazole resistance gene cassettes [153]. In a research study done by Wan et al., the CRISPRi system was developed to target the AcrAB-TolC efflux pump to prevent multi-drug resistance in *E. coli*. It was found that when *E. coli* strains were engineered with plasmids encoding for guide RNA that targeted the efflux pump genes, the CRISPRi system inhibited the transcription of *acrA*, *acrB*, and *tolC* genes, respectively. The engineered strains of *E. coli* HB101 showed susceptibility to rifampicin, erythromycin, and tetracycline, along with lower rates of biofilm formation [154].

The CRISPR Cas system also proved to help study the roles of different genes that play a role in increasing antibiotic resistance. This was shown via research done by Wang et al., who elucidated the role of *blaOXA-23*, *blaTEM-1D*, and *blaADC-25 genes in conferring imipenem and sulbactam resistance in A. baumannii. Using CRISPR-Cas system single gene, double-gene, and triple-gene mutants were created to understand the role of each gene in acquiring antibiotic resistance. They also found that OxyR is responsible for oxidative stress resistance in A. baumannii by exploring the stress-sensing mechanism using the CRISPR-Cas9 system. This suggests that OxyR can be used as a potential drug target in developing novel therapeutics* [155]. *Similarly, the CRISPR Cas9 genome editing tool was also used to study the pan-drug resistance mechanism in K. pneumoniae. In this study, Sun et al., targeted various genes using CRISPR-Cas9 to elucidate their role in conferring resistance against tigecycline and colistin in carbapenem-resistance K. pneumoniae. It was found that knocking out the tetA gene resulted in a decrease in MIC for tigecycline, whereas inactivation of the mgrB gene led to the activation of the* PhoPQ two-component system leading to an increase in MIC for colistin [156]. *E. coli* is responsible for causing catheter-associated UTI outbreaks due to the formation of biofilms in the catheter. Kang et al., showed the reduction in biofilm formation and down-regulation of biofilm-related genes in *E. coli* SE15 isolated from urinary catheters of patients. This was done by incorporating a CRISPR-Cas9 plasmid that targeted the *luxS* gene involved in the quorum sensing mechanism in the wild-type *E. coli* strain. It was concluded that the CRISPR-Cas system can also be used to study mechanisms involved in the autoinducer-2-dependent quorum sensing in other clinically significant bacteria [157].

Newer approaches such as engineering bacteriophages with DNA that encode Cas9 and guide RNA and removing all phage sequences responsible for their replication are currently being explored. On infection of bacteria with these bacteriophages, the CRISPR-Cas9 system will cleave and degrade the bacterial DNA. Bacteria might still develop resistance to such mechanisms, in that case, bacterial mutations will have to be studied and phages will have to be modified to target those specific mutations using the same approach [158].

In this way, CRISPR Cas systems can be used to eliminate either the drug-resistant carrying genes from the bacteria or it can be targeted to kill the AMR pathogen. As evolution of bacterial resistance is unavoidable, different approaches of using the CRISPR Cas system can be assessed further as the system is easily reprogrammable and feasible.

## 6. Challenges of CRISPR Cas Technology

Along with favorable outcomes, other hurdles involved in the use of the CRISPR-Cas system must be explored and studied. Identifying an appropriate delivery system is important to eliminating AMR genes from the pathogen as well as from the community. The delivery system should be assessed and carefully studied such that any perturbation to the native species in the community is avoided. Bacteria will continue to evolve depending upon the different barriers that they will come across. This evolution will give rise to resistance which has to be continually monitored. Strategies designed using the CRISPR Cas system should be such that there are no “escaper mutants” generated that will later hinder the efficiency of the technique. Furthermore, challenges to delivering the CRISPR Cas system against intracellular pathogens need to be explored and studied more. This is because, within the human body, pathogens stay hidden either by encapsulation or by hiding in spaces between tissues. This will limit the efficacy of the CRISPR Cas system where after the treatment, pathogens which survived will again cause the disease.

Some of the major challenges of the CRISPR Cas technology include complexity of microbial communities, delivery mechanisms, resistance to CRISPR Cas system and regulatory approval, which are discussed below.

### 6.1. Complexity of Microbial Communities

The microbial community that exists in the environment, within animals or humans is quite diverse and complex. These natural communities make a part of various microbiomes consisting of more than a thousand species of billions of bacterial cells present in per gram sample of the matrix. This kind of complexity can prove to be a barrier for the CRISPR Cas system when used for the treatment of AMR. Although the CRISPR Cas system has shown immense potential in targeting bacteria or re-sensitizing AMR bacteria to antibiotics, currently, all studies performed have assessed the action of the CRISPR Cas system only in near clonal bacterial populations [159]. Very few in vivo studies using mouse models are done to target a Gram-negative bacterial pathogen specifically to prevent their colonization in the gut [145]. Moreover, in such a microbial population, within a particular species or strain, various kinds of mobile genetics elements (MGE) possessing different resistance genes will be present. Hence, before targeting resistance genes on MGE, the type of resistance gene carriage in bacterial hosts within a microbial population must be determined using time-consuming approaches.

Another challenge is predicting the community response within microbial populations following treatment with the CRISPR Cas system such as the removal of a strain from a population that may cause dysbiosis and allow the growth of unwanted bacteria within the community. Thus, the consequences of AMR removal from complex microbial communities need to be assessed before the application of the CRISPR Cas system [160].

The next challenge is to ensure the spread of the CRISPR Cas system to target bacterial species within native populations which can be hampered because of the presence of unanticipated factors carried by native target bacteria that might affect the conjugation efficiency of the recipients. This is because AMR targeting CRISPR Cas platforms are deployed and assessed using laboratory strains or from clinically isolated strains which may differ from the native strains that are hidden or encapsulated in the community. Consequently, target bacteria cannot be manipulated within a diverse population and hence robust conjugation systems of donors might prove to be less efficient in a complex environment where target bacteria may not be exposed easily [161].

### 6.2. Delivery Mechanisms

CRISPR Cas system is delivered into the target cell using two basic delivery systems which include the viral vectors and the non-viral vectors. Viral vector methods employ the use of phages for the delivery of the CRISPR Cas system into target bacterial cells. However, the efficiency and safety issues of using phages to deliver CRISPR Cas system need to be assessed because hijacked transduction by phages can lead to the delivery of mobile genetic elements and hence, can cause the spread of virulence genes [162]. Consequently, narrow host ranges with widespread distribution of AMR will hinder the use of phages and hence, engineered phages are considered. Furthermore, before applying engineered phages, basic challenges for targeting bacteria in a complex environment within the host need to be studied and addressed such as their specificity, phage pharmacokinetics, encounter rates with target bacteria in complex communities, phage resistance, and entry into animal cell to target intracellular pathogens [159,160,163].

Apart from viral vectors, non-viral vectors using conjugative plasmids have shown potential in the delivery of the CRISPR Cas system. Many in vitro studies have been conducted to show the delivery of the CRISPR Cas system using a suitable bacterial donor. However, there are a few limitations to using conjugative plasmids such as the presence of a restriction-modification system in a recipient, low conjugation efficiency of recipient bacteria, limited host range, presence of barriers in plasmid intake, and external factors in a natural environment [160].

### 6.3. Resistance to CRISPR Cas System

The evolution of bacterial resistance to CRISPR Cas antimicrobials is increasing. According to Uribe et al., both acquired and intrinsic resistance is observed in vitro in bacteria on treatment with CRISPR Cas antimicrobials. This is due to different parameters that need to be carefully studied and applied for the successful killing or re-sensitization of target bacteria. It was observed that low or mild expression of CRISPR Cas9 in *E*. *coli* strains can lead to the escape of survivors by overcoming CRISPR using Rec-A mediated repair. It was observed that mutations in the guide RNA target sequence can also lead to resistance. These mutations can be single nucleotide mutations that may be created due to transposons or insertion elements or can be spontaneous [33]. Furthermore, apart from mutation, anti-CRISPR (*acr*) genes have been identified that prevent the action of the CRISPR Cas system by interfering with their mechanism and allowing the target DNA sequence to escape degradation. These genes were first discovered in *Pseudomonas* phages; at present, twenty-one families *acr* genes have been identified which can act against both type I and type II CRISPR Cas systems and can be transferred between bacterial populations by mobile genetic elements and bacteriophages [159,164]. Such anti-CRISPR proteins can inhibit the activity of CRISPR Cas antimicrobial as some of these genes are also found in virulent strains of *Pseudomonas aeruginosa* and can be transferred to other *P. aeruginosa* strains via conjugation. This will thus hinder the potential of using CRISPR Cas against pathogenic bacterial strains [165].

### 6.4. Regulatory Approval

When considering the use of the CRISPR Cas system for targeting pathogens via horizontal gene transfer (conjugative plasmids), public opinion and acceptance of such systems will always be controversial as this involves the spread of transgene throughout native populations. Similarly, the use of the CRISPR Cas system in clinical settings needs to be thoroughly scrutinized and ought to face regulatory challenges due to its novelty [161].

## 7. Approaches to Overcome Challenges of CRISPR Cas Technology

Many challenges arise when the CRISPR Cas system is used as a tool to eliminate the problem of AMR. Thus, to overcome these challenges, many approaches have been considered and tried. Addressing the issue of delivery of CRISPR Cas using conjugative plasmids can be improved to a certain extent by screening for more conjugative plasmids with an increased host range. When targeting pathogenic bacteria in the gut, Neil et al., screened for conjugative plasmids found in the Enterobacteriaceae family and identified plasmids with high DNA transfer efficiency in the gut microbiota. Following this, Neil et al. developed a genetically engineered probiotic strain for delivery of CRISPR Cas system using high-efficiency plasmid to eliminate chloramphenicol-resistant *E. coli* strains in the mouse gut [166]. However, conjugation is found to be less efficient as compared to phage delivery but can serve as a better approach in eradicating resistant strains or their resistant genetic trait as some conjugative plasmids have a broad host range and can replicate. Furthermore, there is a need to engineer such delivery systems which have a broad host range that will allow us to tackle the use of the CRISPR Cas system in complex microbial communities. The response from bacterial communities to the application is unpredictable and therefore the ecological consequences of the CRISPR Cas system need to be monitored and studied carefully to determine the removal of drug-resistant genes on the frequency of other bacterial species in the population.

To overcome the limitations of available phages for the delivery of CRISPR Cas antimicrobial, phage engineering is the ultimate option. Nath et al., who developed a modified bacteriophage in which the CRISPR Cas system was incorporated into the phage genome, demonstrated this. This system provides two advantages:(A)specific binding of the phage to target bacteria leads to pathogenic apoptosis due to phage infection (abortive infection system)(B)the delivery of the CRISPR Cas system into the targeted pathogen leads to the elimination of the target gene as well as the apoptosis of the pathogenic bacteria [159].

This system was used by Shen et al., to eliminate MDR genes from *K. pneumoniae* by engineering the *Klebsiella* virulent bacteriophage phiKpS2 [167].

However, such phage transferable CRISPR Cas system has slight similarity with phage therapy and thus, will face similar types of obstacles. Second, using CRISPR Cas-based phage system to re-sensitize drug-resistant bacteria on treatment with antibiotics will again counter-select for escaped mutants that are resistant to the treated antibiotics. Therefore, to curb the threat of antibiotic-resistant bacteria and the above limitations, Yosef et al., developed a technology that used temperate phages to deliver the CRISPR Cas system into bacteria. The CRISPR spacers used in this strategy were designed to target antibiotic-resistant genes and genes of the lytic phages. When this CRISPR Cas phage system was introduced into the samples, lysogenic bacteria were re-sensitized to antibiotics on treatment with the drug whereas non-lysogenic bacteria were killed by lytic phages [149]. Consequently, as pathogenic bacteria evolve and might gain resistance to engineered phages, a cocktail of different phages can be created to prevent the escape of resistant mutants and improve the efficiency of the therapy [168].

On the other hand, the anti-CRISPR activity of *acr* genes can be tackled by using multiple variants of the CRISPR Cas system that will overcome the activity of anti-CRISPR proteins. Likewise, Acr-insensitive CRISPR variants can also be engineered to circumvent the issue of anti-CRISPR proteins. Similarly, other approaches such as the use of different Cas proteins apart from Cas9 can be chosen to improve the efficacy and toxicity of Cas9 nucleases in different bacterial hosts [160].

Lastly, as antibiotic resistance is increasing along with failure in conventional control measures, there is an urgent need of developing genetic interventions along with conventional tools to tackle the challenge of antibiotic resistance. Additional challenges of navigating through regulatory approval for using the CRISPR Cas system in the real environment might be overcome with the help of recent technologies [161].

## 8. Conclusions

The increase and spread of drug-resistant pathogens call for an urgent need to develop alternative antimicrobials other than antibiotics. Extensive research has provided promising outcomes and one such result is the use of the CRISPR-Cas system for the detection and removal of bacterial pathogens from an infected body. CRISPR-Cas system has advanced over the years and its mechanism is exploited and modified with pre-existing tools for various applications in the medical field. This system is promising as an antimicrobial tool because of its specificity to target virulence and drug-resistant genes (plasmid and chromosomally encoded) in pathogenic bacteria without affecting other bacterial populations. Moreover, apart from their use in AMR treatment, the CRISPR-Cas system is coupled with other tools and is used for easier, specific, and sensitive detection of bacterial pathogens from samples. Along with this, the detection tools should be made more cost-effective to maximize their use in healthcare systems for early diagnosis.

Different strategies need to be developed to improve efficacy and safety when targeting the pathogen in vivo. Apart from in vitro studies, more in vivo studies using different model systems need to be done to determine other barriers and improve the efficiency of the treatment. This will provide us with data to improve upon its efficacy for in vivo models and will reduce the risk of probable off-target results. Genetically modified phages which have been evaluated for their use as the delivery system should be checked for their efficiency in reaching the affected site and delivering the CRISPR payload to the pathogens present. Newer approaches using CRISPR Cas technology should continue to expand so that existing barriers can be improved with the help of researchers from different fields and modern genetic engineering tools.

## Figures and Tables

**Figure 1 pharmaceuticals-15-01498-f001:**
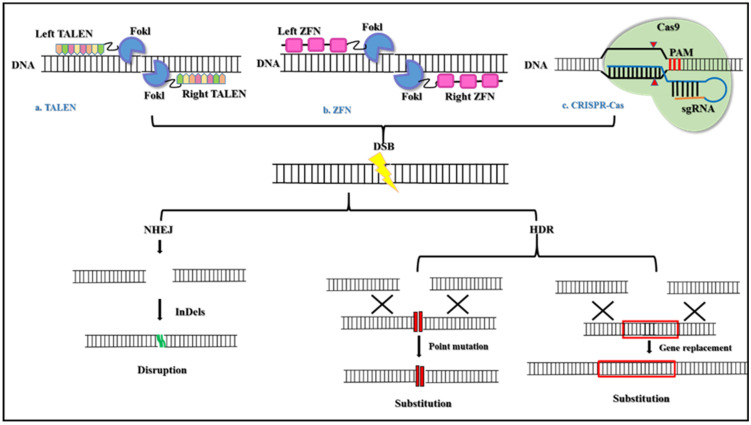
The basic mechanism of Genome editing tools. (**a**) TALEN is engineered endonuclease that are created by fusing a DNA-binding domain obtained from transcription activator-like effectors (TALEs) from *Xanthomonas* proteobacteria and the catalytic domain of FokI endonuclease. The TALEs consist of monomeric tandem repeats of amino acids that bind individually to each nucleotide in the target sequence. (**b**) ZFN has also engineered endonucleases that are generated by fusing the DNA-binding domain of zinc-finger proteins and the catalytic domain of FokI endonuclease. ZFN consists of three to six Cys2-His2 fingers that individually recognize a triplet codon. Both TALEN and ZFN on binding to DNA dimerize to cleave the DNA. (**c**) The CRISPR Cas systems are RNA-guided nucleases found in bacteria that cleave the DNA via RNA-DNA base-pairing. All three tools, TALEN, ZFN, and CRISPR Cas produced double-stranded breaks (DSBs) that are repaired by either non-homologous end joining (NHEJ) or homology-directed repair (HDR) to produce targeted mutagenesis and/or targeted gene substitution [25].

**Figure 2 pharmaceuticals-15-01498-f002:**
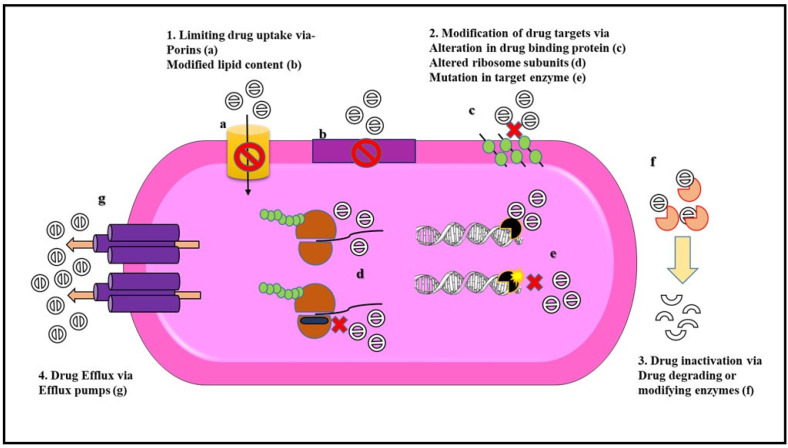
Mechanisms of antibiotic resistance in Gram-negative bacteria. Gram-negative bacteria exhibit different modes of resistance to overcome the action of antibiotics either due to chromosomal mutation or by the acquisition of mobile genetic elements. Some of these resistance mechanisms include (**1**) Limiting drug uptake with the help of (**a**) porins channel that prevents the entry of drug molecule and (**b**) modification of membrane lipid content that acts as a barrier toward the penetration of antibiotics. (**2**) Modification of drug targets either by (**c**) alteration in the drug-binding proteins (e.g., penicillin-binding proteins), or (**d**) altering the target ribosomal subunits (e.g., alteration in the domain V region in the 50S ribosomal subunit leads to linezolid resistance) or (**e**) by a mutation in target enzyme (e.g., mutation of DNA gyrase in fluoroquinolone resistance) which leads to a decrease in the binding affinity of the antibiotic to the target sites. (**3**) Drug inactivation can be either due to (**f**) the production of drug-degrading enzymes such as beta-lactamases or by drug modification via drug-modifying enzymes that conduct biochemical reactions like acetylation phosphorylation and adenylation. (**4**) Drug efflux is exhibited by (**g**) overexpression of efflux pumps that carry out extrusion of antibiotics from the bacterial cell (e.g., AcrAB-TolC in *Enterobacteriaceae*) [58].

**Figure 3 pharmaceuticals-15-01498-f003:**
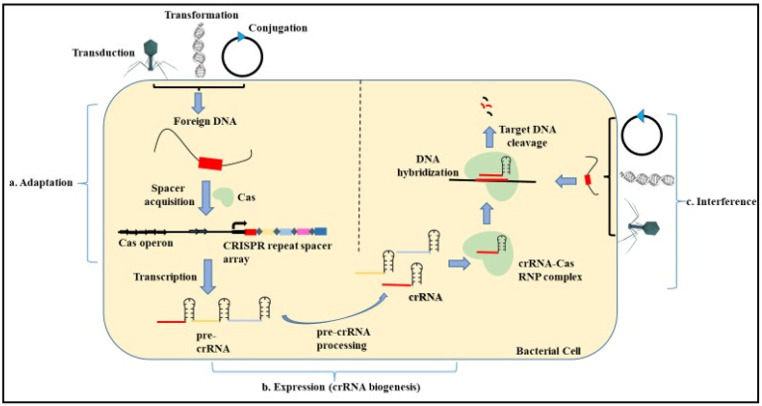
Mechanism of microbial CRISPR Cas System. The mechanism of the CRISPR Cas system involves three main steps. (**a**) Adaptation. In this step, the exogenous DNA is cleaved, followed by the recognition of the proto-spacer adjacent motif (PAM). This proto-spacer is then processed into a pre-spacer having the last PAM nucleotide. The leader end repeat sequence in the CRISPR locus is cleaved followed by the pre-spacer being integrated along with duplication of the repeats flanking the spacer. (**b**) Expression. This step involves the biogenesis of the CRISPR RNA (crRNA) by the transcription of the CRISPR locus containing the spacer sequence. First, the primary CRISPR transcript called the pre-crRNA is generated. These transcripts are further processed by different proteins to give rise to a mature crRNA that contains spacer sequences flanked by partial repeats. (**c**) Interference. In this step, the mature crRNA complexed with the Cas nucleases guides the Cas machinery to the complementary sequences present on the invading nucleic acid. The binding of the crRNA to the homologous sequences leads to the cleavage of the foreign DNA.

**Table 1 pharmaceuticals-15-01498-t001:** Three major categories of drug-resistant pathogenic bacteria as per the CDC records. (Antibiotic Resistance Threats in the United States, 2019 https://www.cdc.gov/drugresistance/biggest-threats.html) [2].

Urgent Threats	Serious Threats	Concerning Threats
Carbapenem-resistant *Acinetobacter* *Clostridioides difficile* (*C. difficile*) Carbapenem-resistant Enterobacteriaceae (CRE) Drug-resistant *Neisseria gonorrhoeae* (*N. gonorrhoeae*)	Drug-resistant *Campylobacter* Extended-spectrum beta-lactamase (ESBL)-producing Enterobacteriaceae Vancomycin-resistant *Enterococci* (VRE) Multidrug-resistant *Pseudomonas aeruginosa* (*P. aeruginosa*) Drug-resistant non-typhoidal *Salmonella* Drug-resistant *Salmonella* serotype Typhi Drug-resistant *Shigella* Methicillin-resistant *Staphylococcus aureus* (MRSA) Drug-resistant *Streptococcus pneumoniae* (*S. pneumoniae*) Drug-resistant Tuberculosis (TB)	Erythromycin-resistant group A *Streptococcus* Clindamycin-resistant group B *Streptococcus*

**Table 2 pharmaceuticals-15-01498-t002:** Antibiotic resistance mechanism employed by major Gram-negative bacteria (Adapted from Ruppé et al., 2015, [41] and modified).

*Enterobacteriaceae*	*Pseudomonas aeruginosa*	*Acinetobacter baumannii*
High-level expressed AmpC cephalosporinase	High-level expressed AmpC cephalosporinase	High-level expressed AmpC cephalosporinase
		High-level expressed OXA-51-like beta-lactamase
Other beta-lactamases	Other beta-lactamases	Other beta-lactamases
Extended-spectrum beta-lactamases	Penicillinases	Extended-spectrum beta-lactamases
Metallo-beta-lactamases (carbapenemases)	Extended-spectrum beta-lactamases	Metallo-beta-lactamases (carbapenemases)
Oxacillinase	Metallo-beta-lactamases (carbapenemases)	Oxacillinase-type carbapenemases
Defect in porins (mutation or impermeability or reduced expression)	Loss of OprD (impermeability)	Functional loss of porins (impermeability). Altered penicillin-binding proteins
Active efflux pumps	Active efflux pumps	Active efflux pumps
OqxAB	MexAB-OprM	AdeABC
AcrAB-TolC	MexXY-OprM	AdeM
QepA	MexEF-OprN MexCD-OprJ	AdeIJK
Aminoglycoside-modifying enzymes	Aminoglycoside-modifying enzymes	Aminoglycoside-modifying enzymes
16S rRNA methylases	16S rRNA methylases	16S rRNA methylases
Topoisomerases modifications	Topoisomerases modifications	Topoisomerases modifications
Lipid A (LPS) modifications	Lipid A (LPS) modifications	Lipid A (LPS) modifications

**Table 3 pharmaceuticals-15-01498-t003:** Classification and characterization of CRISPR Cas system [104,105,106,107].

Class	Type	Spacer Integration Cas Proteins	Pre-crRNA Processing Proteins	crRNA-RNP Complex Proteins	Ancillary Protein	Target Molecule	Cleavage Details
Class 1 (Multi- subunit)	Type I (A-G)	Cas1, Cas2, Cas4	Cas6	Cas11, Cas7, Cas5, Cas8a	Unknown	DNA	Cleaves ssDNA
	Type III (A-F)	Cas1, Cas2		Cas11, Cas7/Csm3, Cas5/Csm4, Cas10, Csm2, Cas7/Csm5	CARF	DNA/RNA	Binds and cleaves nascent RNA
	Type IV (A-C)	Cas1, Cas2		Cas11, Cas7/Csf2, Cas5, Cas8-like Csf1	DinG	Unknown	Unknown
Class 2 (Single-subunit)	Type II (A-C)	Cas1, Cas2, Cas4	RNAse III	Cas9	Csn2	DNA	Blunt-ended dsDNA cleavage
	Type V (A-I, K, U)	Cas1, Cas2, Cas4	Cpf1	Cas12	Unknown	DNA	Staggered DNA dsDNA cleavage
	Type VI (A-D)	Cas1, Cas2	Unknown	Cas13	Unknown	RNA	RNA guided ssRNA cleavage

## Data Availability

Data is contained within the article.

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
