# Peer review of "CRISPR-Cas System: A Tool to Eliminate Drug-Resistant Gram-Negative Bacteria"

_pharmaceuticals, 2022, doi:10.3390/ph15121498_

Round 1

Reviewer 1 Report

This study is focused in discussed the Rapidly emerging drug-resistant superbugs, especially, Gram-negative bacteria, that pose a serious threat to healthcare systems all over the globe. Newer strategies, such as the clustered, regularly interspaced short palindromic repeats/CRISPR-associated protein (CRISPR/Cas) systems, have been proposed for curbing drug-resistant bacteria.

The manuscript is relevant and of great interest. However, the inclusion of more recent bibliography information that supports this revision is required. Continuing, carrying out a more detailed discussion in each section is necessary. The conclusion is very ambiguous and requires critical restructuring. The figures need to be improved substantially. Finally, the reading of the manuscript could be clearer to follow, and authors are recommended to organize the ideas for better understanding.

Author Response

Hello,

Thank you for the meticulous review
As per your suggestions, the manuscript has been revised

Please see the attached PDF

With regards

Reviewer 2 Report

In this manuscript, the author reviewed the progress of drug resistance to pathogens and CRISPR-CAS-associated research. But there are several fatal flaws in this review manuscript as listed below:

1, As claimed in the title, the author should focus this review on applications of the CRISPR-CAS system in eliminating pathogen drug resistance which only was mentioned in a small part of the manuscript with only several references included. The CRISPR-CAS application in drug resistance of bacteria is still mainly focused on the mechanistic study and in-vitro research. The author needs to clearly understand the bottlenecks of CRISPR-CAS application, like delivery method and efficiency, off-target issues, etc., before drafting the manuscript. 

2, it’s curious why the author strengthens the drug resistance of gram-negative bacteria in the manuscript. Both gram-negative and -positive bacterial infection treatments encounter antibiotic-resistance issues and CRISPR-CAS-based application on antibiotic resistance would not distinguish whether the bacteria are gram-negative or -positive. Besides, it seems that the author doesn’t clearly know which pathogens are gram-negative or -positive. Many cases cited in the review were associated with gram-positive pathogens, like C.diff, S. aureus, E. fecalis, M. tub, etc. However, it’s claimed that only gram-negative bacteria were focused on in this review manuscript.

3, it’s not clear why the author put drug resistance of bacterial pathogen and cancer treatment in one review, I fail to see a link strong enough to connect them.

Author Response

(The authors gave the same response as above.)

Reviewer 3 Report

The authors describe a review about the application of CRISPR-Cas system in managing the drug-resistant gram-negative bacteria. While to be consistent with the focus on the Special Issue theme, information about the application of the technique on cancer is not suitable or necessary to be included in the review. Therefore, the word cancer (in the title) and information on point 6 in page 12 as well as in the conclusion should be removed.

Other comments are as follows:

1. If we refer to bacteria group, it should be used small letter of 'g' for gram-negative bacteria if it mentioned in the middle of the sentence. However, if we refer to the technique used to classify the bacterial group, then the capital of 'G' is used to refer the Gram staining method for classification of the Gram positive and Gram negative.

2. Revise and write an appropriate title for Table 1.

3. Subtopic 2.1. The information is not necessary as it can be found better in microbiology textbook. Simplified the characteristics of GNB in a short paragraph.

4. Subtopic 2.2. What is the characteristics of the OM of the gram-negative bacteria that make it more resistant than gram-positive bacteria. Add the information after the 1st sentence.

5. Enterobacteriaceae is a family name. Therefore, it should not italicized. It is same goes to Order name (Enterobacterales), should not italicized. Please also check for other family name of bacteria in the text.

Please check other comments/typo as stated in the reviewed manuscript.

Author Response

(The authors gave the same response as above.)

Reviewer 4 Report

In this manuscript, a review of the CRISPR-Cas system for the elimination of Gram-negative bacteria resistant to antibiotics is made. Its use in the diagnosis and treatment of cancer is also mentioned.

The following comments are made:

1. Do not use the abbreviation GNB. The correct term is Gram negative bacteria. Correct in all the text

2. Figure 1. Put what DSB, NHEJ and HDR mean, in the figure caption.

3. “eliminate AMR Gram-negative”. Put what AMR means

4. “an endotoxin that Other applications”. What do you refer to? Not clear, correct

5. “GNB are intrinsically more resistant than Gram-positive”. But Gram-positives have a thicker peptidoglycan layer that makes them more resistant.

6. “such as P.aeruginosa PAO1, E.cloacae ATCC 13047”. Correct font size

7. Put the meaning of MIC

8. Explain briefly Figures 2 and 3 in the footer

9. “TMP-SMZ”. put what it means

10. “Wang took a similar approach et al.” Correct: Wang et al.

11. Sections 5.2, 5.3, 6.1 and 6.2 divide into several paragraphs for better reading.

12. “sgRNA are expressed”. Put what sgRNA means and explain what it is and how it is used. important part of the system.

13. “Using RNA-guided nucleases such as the CRISPR-Cas system the virulence factor eae gene in enterohemorrhagic E. coli O157:H7…..”. E. coli is Gram negative, why do you put it in another section? Put it in the Gram negative section

14. In my opinion, putting the detection and treatment of cancer out of the context of the general theme of the manuscript, which is Gram negative bacteria. Since you would have to have the whole context of cancer and it would make a very confusing manuscript.

Author Response

(The authors gave the same response as above.)

Round 2

Reviewer 1 Report

None comment

Author Response

Thank you, Sir, for accepting the answers to your comments of round 1.

With regards

Reviewer 2 Report

I have no further questions

Author Response

Thank you, Sir

Reviewer 4 Report

The authors made the suggested changes however the following comments are made:

1. “especially gram-negative bacteria”. Gram is a proper noun, capitalize. Correct throughout the text.

2. The conclusion seems part of the Discussion. Make a more concrete conclusion.

Author Response

Thank you for the round 2 suggestions.

My replies to the comments

The authors made the suggested changes however the following comments
are made:

1. “especially gram-negative bacteria”. Gram is a proper noun,
capitalize. Correct throughout the text. - We have made corrections throughout the text.

2. The conclusion seems part of the Discussion. Make a more concrete
conclusion. - The conclusion is revised.

With Regards